# Identification of the Extracellular Nuclease Influencing Soaking RNA Interference Efficiency in *Bursaphelenchus xylophilus*

**DOI:** 10.3390/ijms232012278

**Published:** 2022-10-14

**Authors:** Ruijiong Wang, Yongxia Li, Dongzhen Li, Wei Zhang, Xuan Wang, Xiaojian Wen, Zhenkai Liu, Yuqian Feng, Xingyao Zhang

**Affiliations:** 1Key Laboratory of Forest Protection of National Forestry and Grassland Administration, Ecology and Nature Conservation Institute, Chinese Academy of Forestry, Beijing l00091, China; 2Co-Innovation Center for Sustainable Forestry in Southern China, Nanjing Forestry University, Nanjing 210037, China

**Keywords:** *Bursaphelenchus xylophilus*, RNAi efficiency, dsRNA stability, extracellular nuclease

## Abstract

RNA interference (RNAi) efficiency dramatically varies among different nematodes, which impacts research on their gene function and pest control. *Bursaphelenchus xylophilus* is a pine wood nematode in which RNAi-mediated gene silencing has unstable interference efficiency through soaking in dsRNA solutions, the factors of which remain unknown. Using agarose gel electrophoresis, we found that dsRNA can be degraded by nematode secretions in the soaking system which is responsible for the low RNAi efficiency. Based on the previously published genome and secretome data of *B. xylophilus*, 154 nucleases were screened including 11 extracellular nucleases which are potential factors reducing RNAi efficacy. To confirm the function of nucleases in RNAi efficiency, eight extracellular nuclease genes (*BxyNuc1*-*8*) were cloned in the genome. *BxyNuc4*, *BxyNuc6* and *BxyNuc7* can be upregulated in response to ds*GFP*, considered as the major nuclease performing dsRNA degradation. After soaking with the dsRNA of nucleases *BxyNuc4*/*BxyNuc6*/*BxyNuc7* and *Pat10* gene (ineffective in RNAi) simultaneously for 24 h, the expression of *Pat10* gene decreased by 23.25%, 26.05% and 11.29%, respectively. With soaking for 36 h, the expression of *Pat10* gene decreased by 43.25% and 33.25% in ds*BxyNuc6*+ds*Pat10* and ds*BxyNuc7*+ds*Pat10* groups, respectively. However, without ds*P**at10*, ds*BxyNuc7* alone could cause downregulation of *Pat10* gene expression, while ds*BxyNuc6* could not disturb this gene. In conclusion, the nuclease *BxyNuc6* might be a major barrier to the RNAi efficiency in *B. xylophilus*.

## 1. Introduction

The pine wood nematode, *Bursaphelenchus xylophilus*, is a notorious, invasive, plant-parasitic nematode that causes pine wilt disease [1], which is the most complex and devastating disease in forest ecosystems in China, South Korea, Japan, Portugal and other European countries [2,3,4]. It has caused significant economic and ecological damage in China since it was first reported in Nanjing in 1982 [5]. Pine wilt disease has spread rapidly to 19 provinces including 731 county-level administrative regions in China [6]. Upon invasion of a pine tree, *B. xylophilus* can reproduce quickly and destroy the vascular system of the entire tree, causing wilting and death in only a few weeks [7]. The main control measures include trunk injection of nematicide and eradication of the damaged trees and vector insects. However, large-scale application of pesticides has led to environmental issues and health concerns [8]. Therefore, development of an efficient and environmentally friendly approach to control *B. xylophilus* is urgently needed. RNA interference (RNAi) is a cellular mechanism in which double-stranded RNA (dsRNA) molecules drive the post-transcriptional silencing of genes with homologous sequences [9]. Since the RNAi response was first described in the free-living nematode *Caenorhabditis elegans* [10], it has become a valuable tool for studying gene function of nematodes, and it may be a promising approach for the control of plant-parasitic nematodes [11]. The interference of plant-parasitic nematodes is mostly achieved by soaking [12,13]. However, the efficiency of this approach varies widely with the nematode species [14]. In *B. xylophilus*, the RNAi efficiency ranged from 30% to 80%, while some studies showed no effect [15,16,17,18]. Even when targeting the same gene, the experimental results of different studies were inconsistent. For example, the mortality rate of pine wood nematode is up to 80% after interference with *BxAK1* for 8 h [19]. However, in another report, the nematodes did not die or show the expected phenotype changes by feeding yeast cells, which can express dsRNA to interfere with AK1 and AK2 (iGEM. http://2018igemorg/Team:Kyoto/Experiments (accessed on 15 November 2021)).

Efficient RNAi-induced gene silencing requires some essential processes, including delivery of dsRNA, uptake from the hemolymph or gut, dsRNA processing by RNAi enzymes, intracellular transport and expression of the core RNAi machinery [20]. Above all, the stability of dsRNA is the first step to influencing RNAi efficiency before uptake. When ingested, dsRNA should avoid dsRNase degradation and move from intestinal or hemolymph lumen to tissues to effectively exert its inhibitory effect. In recent years, many studies have suggested that extracellular nuclease is the important factor leading to low RNAi efficiency in insects, such as *Ostrinia furnacalis* [21] and *Drosophila suzukii* [22]. When silencing the extracellular nuclease or inhibiting the activity of enzymes, RNAi efficiency is significantly improved. However, different insects have different types and activities of extracellular nucleases in different physiological states, which may be an important factor affecting the difference and instability of RNAi efficiency in insects [23,24]. Plant-parasitic nematodes secrete a variety of functional enzymes during their feeding process to digest cellular components and resist plant defense responses, among which extracellular nucleases play an important biological role in the absorption of nutrients and removal of alien substances [25,26]. At present, the function of plant-parasitic nematode extracellular nucleases is scarcely studied, and their effect on RNAi efficiency is still unclear. In the study of pine wood nematodes, there have been few proposed solutions to the problems of low interference efficiency or ineffective interference. 

Through our study we want to draw attention to whether the extracellular nucleases of *B. xylophilus* can degrade dsRNA in the soaking system, affecting the stability of dsRNA before it is taken up by nematode intestinal cells. Therefore, screening key nucleases and clarifying the function of these enzymes in *B. xylophilus* RNAi has theoretical significance for the study of functional genes and production of new control methods for *B. xylophilus.*

## 2. Results

### 2.1. In Vitro Degradation of dsRNA by Nematodes

The ds*GFP* and ds*Pat10* were incubated at 25 °C with nematodes for 24 and 48 h, respectively. After 24 h, dsRNA showed obvious degradation and the intensity of electrophoretic bands of the nematode treatment group of ds*GFP* and ds*Pat10* had significantly reduced by 56.2% and 67.2%, respectively (Figure 1a and Figure 2). After 48 h, the electrophoretic bands of the nematode treatment group were almost invisible (Figure 2). The relative intensity of electrophoretic bands was only 0.081 and 0.073 in ds*GFP* and ds*Pat10* groups, respectively (Figure 1a).

In order to detect whether the secretions of *B. xylophilus* in water degrade dsRNA, the sterilized supernatant of *B. xylophilus* was collected. Compared with the supernatant-heated control group and the water-immersed control group, the unheated supernatant treatment group of ds*GFP* and ds*Pat10* showed significantly weaker electrophoretic bands at 24 and 48 h (Figure 1b and Figure 3). However, not all of them were degraded (Figure 3). The relative intensity of heated supernatant has no significant difference with the water control in ds*GFP* groups for 48 h and ds*Pat10* groups for 48 h (Figure 1b). The degradation effect of the supernatant group was lower than that of the nematode group (Figure 2 and Figure 3). It is suggested that enzymes in the soaking system of *B. xylophilus* could degrade dsRNA, and the presence of nematodes accelerates the degradation of dsRNA. On the one hand, some of the dsRNA was ingested by nematodes; on the other hand, the nematode may secrete more enzymes.

### 2.2. Screening of Extracellular Nuclease Genes

The extracellular nucleases of *B. xylophilus* may contribute to the degradation of dsRNA *in vitro*. To confirm whether or which nucleases secreted by nematodes cause dsRNA degradation, we screened the extracellular nuclease genes of *B. xylophilus*. According to the reported genome database of *B. xylophilus*, 154 nuclease genes were preliminarily screened (Appendix A), and 11 extracellular nuclease genes were screened through signal peptide prediction. Among them, eight genes have been successfully cloned, named *BxyNuc1-8* (Accession numbers: OP482151-OP482158). The comparison table between accession number and gene ID of these eight genes is shown in Appendix A. The ORF length, amino acid sequence length, signal peptide position, isoelectric point and relative molecular mass of these eight genes were analyzed (Table 1). The molecular weight of the encoded proteins was approximately 30–40 kDa. According to the phylogenetic analysis, *BxyNuc*1, *BxyNuc*2, *BxyNuc*3 and *BxyNuc*7 are far from the other nucleases in clustering. The *BxyNuc6* is in a clade with plant-parasitic nematodes and insects, and it is speculated that it may have similar dsRNA degradation function to the reported nuclease of insects. Furthermore, *BxyNuc4*, *BxyNuc5* and *BxyNuc8* are in a clade with mammalian parasitic nematodes (Figure 4).

### 2.3. Extracellular Nuclease Gene Response to dsGFP

In order to screen extracellular nuclease genes in response to dsRNA soaking, the relative mRNA expression levels of the cloned nuclease genes were detected by qRT-PCR after soaking nematodes with non-target ds*GFP.* The results showed that the expression of nuclease *BxyNuc4* and *BxyNuc6* can be significantly upregulated by introduction of dsGFP for 24 h. The expression of *BxyNuc1*, *BxyNuc2* and *BxyNuc3* were significantly downregulated (Figure 5). After soaking for 48 h, the expression of nuclease *BxyNuc7* was significantly upregulated compared with the control (Figure 5). Furthermore, the expression of *BxyNuc4* and *BxyNuc6* were upregulated. Therefore, the nuclease genes *BxyNuc4*, *BxyNuc6* and *BxyNuc7* were used for further experiments and may play a major role in the degradation of dsRNA during the RNAi process of *B. xylophilus.*

### 2.4. The Silencing of Extracellular Nuclease Genes Improves RNAi Efficiency

Multiple experiments showed that the expression of *Pat10* gene was not downregulated after 24 and 36 h of interference (Figure 6a). However, the nuclease *BxyNuc4, BxyNuc6* and *BxyNuc7* can be effectively interfered (Figure 6b). In order to further investigate the contributions of nucleases *BxyNuc4*, *BxyNuc6* and *BxyNuc7* to the inefficiency of RNAi in *B. xylophilus*, we mixed the same concentration of nuclease dsRNA and ds*Pat10* (0.5 µg·µL^−1^) for interference simultaneously. The results showed that the transcript levels of each nuclease gene were significantly reduced for 24 h (Appendix A). At the same time, the relative mRNA expression level of *Pat10* gene was 0.975, 0.827 and 0.855 in ds*BxyNuc4*+ds*Pat10*, ds*BxyNuc6*+ds*Pat10* and ds*BxyNuc7*+ds*Pat10* groups after 24 h of interference, respectively (Figure 7a). It is indicated that after co-interference, *BxyNuc6* and *BxyNuc7* could significantly downregulate the expression of *Pat10* gene. Thus, we choose *BxyNuc6* and *BxyNuc7* to interfere simultaneously with *Pat10* for 36 h. The expression of *Pat10* gene was 0.641 and 0.741 in ds*BxyNuc6*+ds*Pat10* and ds*BxyNuc7*+ds*Pat10* groups, respectively (Figure 7b). It was therefore revealed that the RNAi efficiency of *Pat10* was indeed improved over time with the participation of nucleases *BxyNuc6* or *BxyNuc7*.

Using ds*GFP*+ds*BxyNuc6/Nuc7* as a group to detect the expression of *Pat10* for the purpose of excluding the influence of nuclease dsRNA alone on the interference of *Pat10* gene. Compared with the control (ds*GFP*+ds*GFP*), the expression of *Pat10* was 1.189, 1.056 and 0.757 in ds*GFP*+ds*Pat*10, ds*GFP*+ds*BxyNuc6* and ds*BxyNuc6*+ds*Pat10* groups, respectively (Figure 7b). The results showed that ds*BxyNuc6* alone (without ds*Pat10*) does not cause downregulation of the *Pat10*. However, when the nuclease was changed to *BxyNuc7*, the results showed that the expression of *Pat10* was 0.819 and 0.874 in ds*GFP*+ds*BxyNuc7* and ds*BxyNuc7*+ds*Pat10* groups, respectively (Figure 7b). It is suggested that ds*BxyNuc7* (without ds*Pat10*) could cause the downregulation of *Pat10.* To sum up, the interference efficiency of *Pat10* gene can indeed be improved by interfering with nuclease *BxyNuc6* or *BxyNuc7.* However, only the *BxyNuc6* can directly affect RNAi efficiency, and the interference of *BxyNuc7* could generate downregulation of *Pat10*, probably because this interference affected the vital movement of nematodes and led to changes in the expression of *Pat10*. This provides a feasible method for the low efficiency of lethal gene interference.

## 3. Discussion

RNAi technology is becoming one of the most promising tools for gene function studies of nematodes. However, RNAi efficiency is relatively low and unstable in plant-parasitic nematodes compared with *C. elegans*. It is known that the efficiency of gene knockdown by RNAi is influenced by several factors including dsRNA degradation, different delivery method of dsRNA, selection of target genes and the susceptibility of organisms to RNAi [15,23,27,28]. Our study found that dsRNA is unstable in the *B. xylophilus* soaking system. The agarose gel electrophoresis results suggest that the nuclease secreted by *B. xylophilus* is probably the major reason for this aspect. Interference with nuclease *BxyNuc6* and *BxyNuc7* significantly reduced the expression of these two nuclease transcripts and substantially improved RNAi efficiency of *Pat10* gene as well (Appendix A and Figure 7a). This provides an important basis for the follow-up research, but it still cannot rule out the possible degradation function of other extracellular nucleases such as *BxyNuc4*. A phylogenetic analysis showed high similarity among the *BxyNuc6* and the nucleases that have been reported to degrade dsRNA in insects [20,22,29,30,31]. Whether the *BxyNuc6* also has the function of degrading dsRNA needs to be further verified with a protein function study.

In insects, numerous studies have revealed that a poor RNAi response is usually associated with high double-stranded RNA (dsRNA)-degrading activity. The oral delivery of dsRNA usually has lower RNAi efficiency compared with microinjection, and the reason for this phenomenon is the exposure of the dsRNA to nucleases secreted in the gut juice and the unsuitable gut pH. Due to the degradation of dsRNA, the uptake of dsRNA by cells is not enough to cause the continuous interference of target genes, which may be the reason for the poor interference effect of this method. Therefore, dsRNA exposure should persist long enough to allow cellular uptake [20]. Furthermore, specifically silencing these nuclease genes can significantly improve the interference efficiency of oral dsRNA. This result was confirmed in multiple insect species, for example, *Tribolium castaneum* [32], *Anthonomus grandis* [20], *Ostrinia furnacalis* [21] and *Aedes aegypti* [33]. Understanding the interaction between the insects’ nuclease activity and dsRNA is expected to improve the application of RNAi technology in pest control, as well as in plant-parasitic nematodes. However, the role of extracellular nucleases in plant-parasitic nematodes remains uncertain. In *C. elegans*, only an *endonuclease* (NM_058970.4) has been reported [34] and seven extracellular nucleases were identified (Appendix A). In this study, a total of 11 extracellular nucleases were identified in *B. xylophilus* and the function of *BxyNuc4*, *BxyNuc6* and *BxyNuc7* may be related to dsRNA degradation in nematodes. It is implied that the *B. xylophilus* has stronger ability of nucleic acid degradation outside the cell. In a leaf beetle (*Plagiodera versicolora*), it is proven that the degradation products of dsRNA can be utilized by gut bacteria for growth [35]. However, in *B. xylophilus*, whether the degraded dsRNA is utilized by nematodes needs to be further explored.

Different extracellular nucleases may have functional complementarity. The downregulation of the target nuclease gene may lead to increased expression of other nucleases. This phenomenon has been demonstrated in the red flour beetle *Tribolium castaneum* [32]. Furthermore, some off-target effects also occur between different nucleases genes [32,36]. In this study, we found some extracellular nucleases were highly expressed in response to ds*GFP,* and the others were significantly downregulated. Whether there is functional complementarity between *BxyNuc4*, *BxyNuc6* and *BxyNuc7* or other extracellular nucleases screened in this study still needs to be confirmed. At the same time, dsRNA is also likely to be degraded in the gut of nematodes, like most insects. Since the successful implementation of the RNAi process requires the participation of multiple links [9,23,37], whether there are other links that affect the interference efficiency is unknown.

In the process of interference, different target genes showed different interference efficiency. For example, in the root-knot nematode (*Meloidogyne incognita*), 20 genes involved in the RNAi pathways were investigated, and two of the genes could not be knocked down. Only 10 of the genes were significantly downregulated. The results showed that the genes may respond to RNAi knockdown differently, so an exhaustive assessment of target genes as targets for nematode control via RNAi is imperative [13]. In *B. xylophilus*, low RNAi efficiency has been described for genes such as *Bx-unc-87*, *Bx-tmy-1* and *Bx-hsp-1* [15,38]. In this study, the *Pat10* gene showed lower RNAi efficiency than that of the nucleases (Figure 6). This may also be related to the gene itself or the location of gene expression [38].

In recent years, achieving RNAi by feeding nematodes fungi which can express dsRNA is a promising method for nematode control [16,39], but the soaking method still has a considerable role. It can preliminarily screen the functions of some important genes of nematodes with relatively simple operation. Therefore, it is more important to improve the interference efficiency of the soaking method. The co-interference of nucleases and target genes has been proved to be an effective method in this study. In addition, additives that alter the enzymatic activity or pH in the soaking system might be a good solution for enhancing RNAi efficiency [23]. In conclusion, our study found that the in vitro protein secretion of *B. xylophilus* can degrade dsRNA in the soaking system, affecting the stability of dsRNA before it is taken up by nematode intestinal cells. Among the nucleases that we screened, *BxyNuc6* might be a major barrier to the RNAi efficiency in *B. xylophilus*. The studies of extracellular nuclease illustrate the importance of delivery method of dsRNA and provide a feasible idea for improving RNAi efficiency.

## 4. Materials and Methods

### 4.1. Cultivation of Nematodes

*Bursaphelenchus xylophilus* NXY61 was isolated from diseased *Pinus massoniana* in Ningbo, Zhejiang Province, China, in May 2015. This nematode has been cultured in the laboratory for generations. Thus, the nematodes we used were a stable genetic laboratory strain. The nematodes were cultured on the mycelia of *Botrytis cinerea* on potato dextrose agar (PDA) plates at 25 °C for 5 days. The nematodes were washed twice with 0.1× PBST buffer and collected in a 15 mL centrifuge tube via centrifugation at 3000 rpm for 30 s. These nematodes were cleaned with sterile water 3~5 times until the supernatant was clear. In order to prevent the influence of bacteria during the experiment, the nematodes were treated with 0.5% sodium hypochlorite for 30 s and cleaned with sterile 1× PBS buffer twice. Finally, the activity of nematodes was observed under the microscope to ensure the nematodes thrived, and nuclease-free water was added for further experiments.

### 4.2. Synthesis of dsRNA

The total RNA was extracted from nematodes using TransZol Up Plus RNA Kit (Transgen Biotech, Beijing, China). cDNA was synthesized from 1 µg of total RNA using the PrimeScript RT reagent Kit with gDNA Eraser (TaKaRa, Beijing, China). The DNA template was prepared for dsRNA synthesis by using PCR with primers designed to add T7 promoter sites at both ends. All primer sequences are listed in Appendix A. Subsequently, dsRNA was synthesized and purified using the T7 RiboMAX^TM^ Express RNAi System (Promega, Madison, USA) according to the manufacturer’s instructions. Integrity of dsRNAs was evaluated by electrophoresis in 1.2% agarose gels, and the amounts of dsRNA were quantified with a spectrophotometer (Nano-Drop 2000, Thermo Scientific, Waltham, USA).

### 4.3. Degradation of dsRNA Detected by Agarose Gel Electrophoresis

Approximately 300 nematodes were incubated with 5 µL of dsRNA (final concentration 0.5 µg·µL^−1^) in a 50 µL volume. Reactions were incubated at 25 °C for 24 and 48 h. The sample was dissolved in an equal volume of nuclease-free water and centrifuged briefly for 30 s. Taking part of the supernatant and add RNA Loading Dye (Takara, Beijing, China), the mixture was heated at 65 °C for 10 min. The dsRNA integrity was analyzed by means of 1.2% agarose gel electrophoresis. Gels were scanned using a fluorescence laser scanner (Azure Biosystems, Dublin, USA).

In addition, approximately 100,000 nematodes were incubated in a shaking incubator at 180 rpm for 48 h at 25 °C. The nematodes were centrifuged at 5000 rpm for 3 min, and the supernatant was sterilized with a 0.22 µm filter. A part of the supernatant was heated at 80 °C for 20 min to inactivate the enzyme. A total of 45 µL of heated and unheated supernatants was incubated with 5 µL of dsRNA (final concentration 0.5 µg·µL^−1^) for 24 and 48 h separately. As a positive control, the same concentration and volume dsRNA was incubated in nuclease-free water. There were three replicates for each treatment. The integrity of the dsRNA was evaluated using the same method as above. The gray values of the electrophoretic bands were measured with Azure Spot software (Azure Biosystems, Dublin, USA).

### 4.4. Screening and Cloning of B. xylophilus Extracellular Nuclease Genes

To obtain an initial set of candidate nucleases in *B. xylophilus*, the amino acid sequences of *Caenorhabditis elegans* (NP_491371.1, NP_492590.1) were used as query sequences to search the *B. xylophilus* transcriptome database (PRJEA64437) using the local BLAST program (blast-2.2.30+). At the same time, the nucleases were retrieved according to the annotation of the *B. xylophilus* genome (PRJEA64437). All screened nucleases were predicted by signal peptide on the SignalP-5.0 Server (http://www.cbs.dtu.dk/services/SignalP/ (accessed on 12 October 2021)) to determine whether they could be secreted out of the cell with the signal peptide. Each retrieved full-length open reading frame (ORF) was confirmed by PCR amplification with the 2× PCR Mix (Tiangen, Beijing, China). The PCR program was as follows: 94 °C for 3 min followed by 37 cycles of 94 °C for 30 s, 55 °C for 30 s and 72 °C for 1 min. The predicted target band was excised and recovered using the QIAEX II Gel Extraction Kit (QIAGEN, Shanghai, China). The samples were subsequently sequenced by the BGI company to verify the nucleotide sequences.

The molecular weight and predicted theoretical isoelectric point (pI) were calculated using the Compute pI/Mw tool (https://web.expasy.org/compute_pi/ (accessed on 12 October 2021)). Using the cloned extracellular nuclease sequences of *B. xylophilus* as query to screen the orthologous proteins (query cover > 70%) in other species utilizing the blastp tool (https://blast.ncbi.nlm.nih.gov/Blast.cgi (accessed on 20 December 2021)). The insect nuclease sequences were derived from articles reporting that these nucleases have the function of dsRNA degradation. These amino acid sequences were used for the construction of a phylogenetic tree using the maximum likelihood method (1000 bootstrap repeats).

### 4.5. Interference of Target Gene and Extracellular Nuclease Genes

The *Caenorhabditis elegans* gene *pat10* is an essential component of the body wall muscle [40] and thus is required for nematode movement. The *B. xylophilus* orthologs of this gene were used in this study. Using the amino acid sequence of *C. elegans pat10* gene (wormbase ID F54C1.7) as the query sequence, the protein sequences with the highest percentage similarity were further researched by searching the published *B. xylophilus* genome in NCBI using tblastn. Sequences with the same alignment results were searched in the dataset of the *B. xylophilus* genome (PRJEA64437) to obtain the cDNA sequence of *Pat10.*

Freshly cultured nematodes of *B. xylophilus* (a mix of adults and juveniles, approximately 10,000 individuals) were immersed in 100 µL solution containing dsRNA (final concentration 1 µg·µL^−1^) and incubated in a shaking incubator at 180 rpm for 24 h/36 h/48 h at 25 °C. The dsRNA solutions which contained the same final concentrations (0.5 µg·µL^−1^) of Pat10 dsRNA and nuclease dsRNA were used for simultaneous interference. Equal numbers of nematodes immersed in GFP dsRNA solution were treated as controls. There were three replicates for each treatment. After soaking, the samples of each treatment were thoroughly washed several times in 0.1× PBST sterile water and then used for further experiments.

### 4.6. Quantitative Reverse Transcription PCR (qRT-PCR)

Prior to use in qRT-PCR, cDNA was 1:4 diluted with ddH_2_O. All primers used were designed with Primer Premier 6. The qPCR reaction was set up in 20 μL containing 1 μL of cDNA, 10 μL of TB Green Premix DimerEraser (Perfect Real Time), 1 μL of each primer and ddH_2_O. cDNA templates were denatured at 95 °C for 30 s, followed by 40 three-segment cycles of amplification at 95 (5 s), 55 (30 s) and 72 °C (30 s). A melting curve analysis was performed after the qPCR run (15 s at 65 °C). The actin gene of *B. xylophilus* was used as the internal control. The experiment had three biological replicates and four technical replicates. According to the cycle threshold (Ct) value and the dissolution curve, the 2^−ΔΔCt^ method was used to estimate the relative expression level of the target gene and verify the interference efficiency.

### 4.7. Statistical Analysis

Statistical significance of differences in the mRNA expression levels among different treatment groups was determined by one-way analysis of variance in PASW Statistics 18.0 software (at *p* < 0.05).

## Figures and Tables

**Figure 1 ijms-23-12278-f001:**
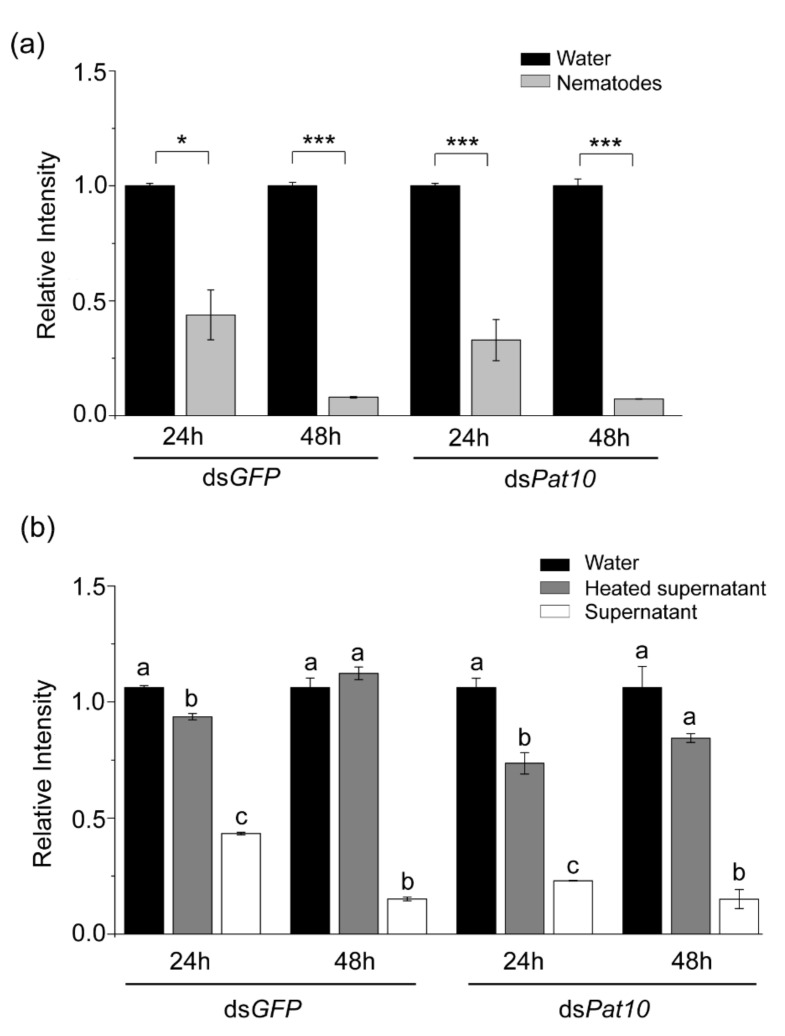
Relative intensity of agarose gel electrophoretic bands. (**a**) The relative intensity of electrophoretic bands after dsRNA and nematodes soaked for 24 and 48 h. (**b**) The relative intensity of electrophoretic bands after dsRNA and nematode secretions soaked for 24 and 48 h. Relative intensity differences between two groups were calculated using the independent samples *t* test. Data are presented as mean ± SE, *** *p* < 0.001; * *p* < 0.05; different letters indicate significant difference (*p* < 0.05).

**Figure 2 ijms-23-12278-f002:**
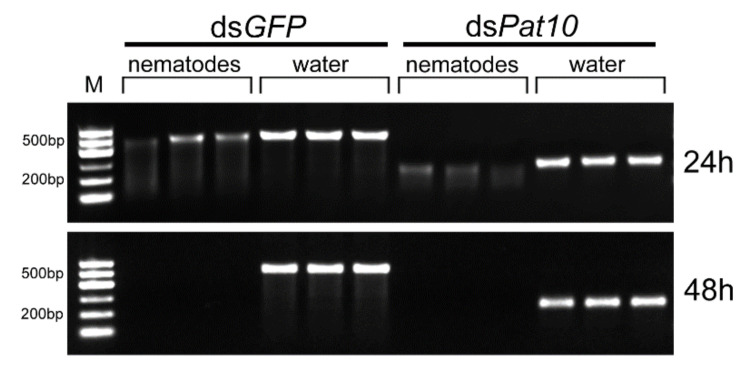
Agarose gel electrophoresis results of dsRNA and nematodes soaked for 24 and 48 h.

**Figure 3 ijms-23-12278-f003:**
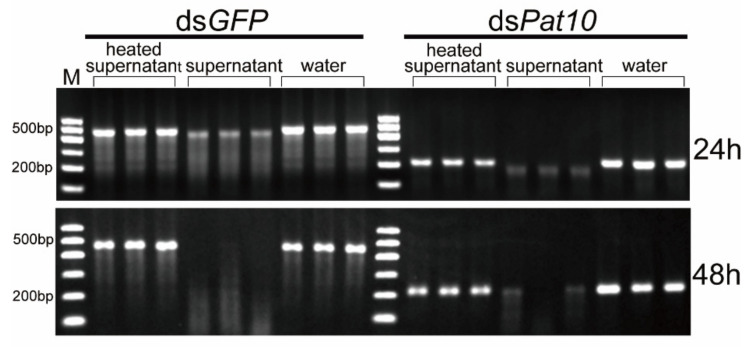
Agarose gel electrophoresis results of dsRNA and nematode secretions soaked for 24 and 48 h.

**Figure 4 ijms-23-12278-f004:**
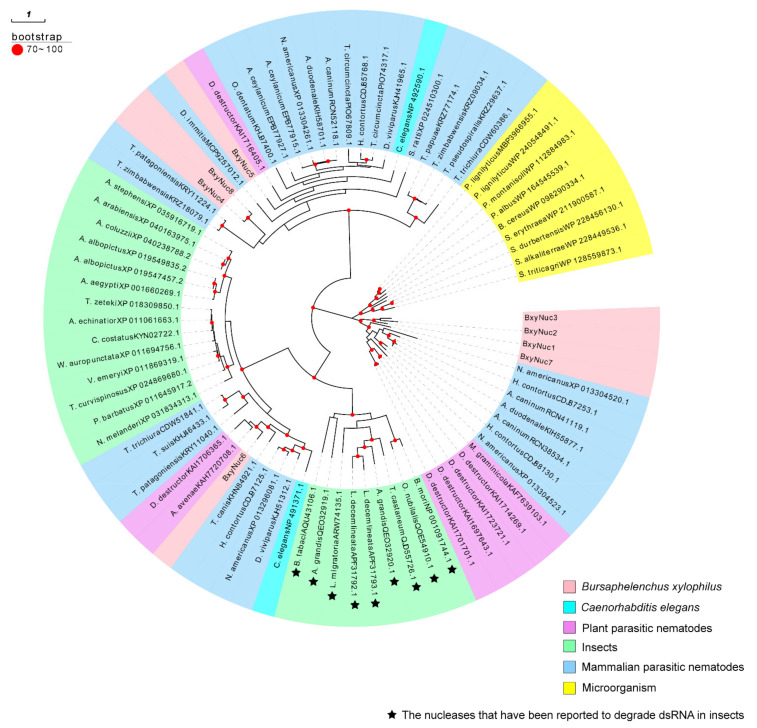
A phylogenetic tree constructed using 75 amino acid sequences from *B. xylophilus, C. elegans*, plant-parasitic nematodes, insects, mammalian parasitic nematodes and microorganisms with the maximum likelihood method (1000 bootstrap repeats).

**Figure 5 ijms-23-12278-f005:**
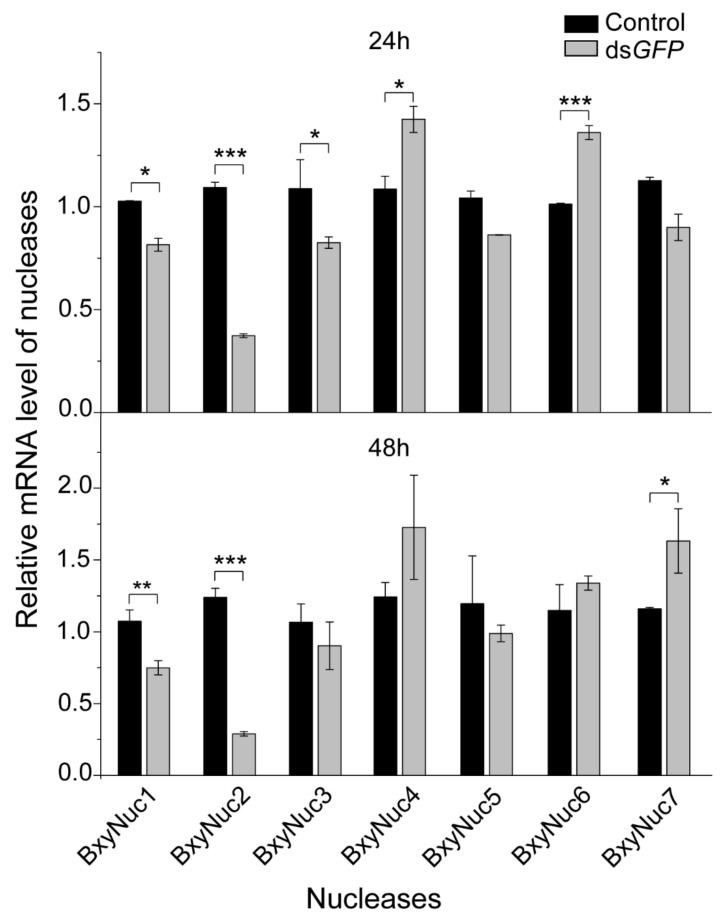
The relative expression level of nuclease genes in nematodes after soaking with ds*GFP* for 24 and 48 h. Gene expression differences between two groups were calculated using the independent samples *t* test. Data are presented as mean ± SE, *** *p* < 0.001; ** *p* < 0.01; * *p* < 0.05.

**Figure 6 ijms-23-12278-f006:**
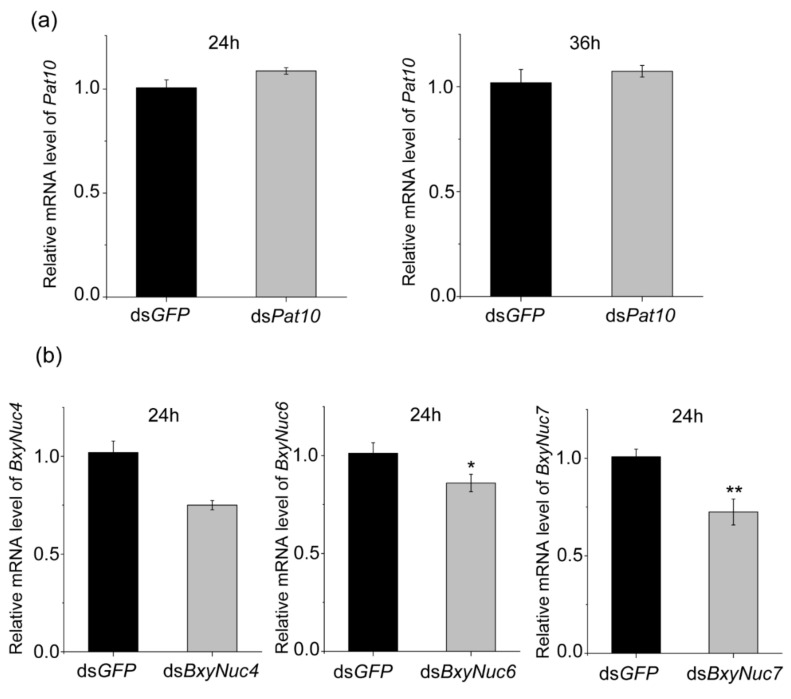
The relative expression level of *Pat10* and nucleases genes. (**a**) The relative mRNA level of *Pat10* after interfering for 24 and 36 h. (**b**) The relative mRNA level of *BxyNuc4*/*BxyNuc6*/*BxyNuc7* after interfering for 24 h. Gene expression differences between two groups were calculated using the independent samples *t* test. Data are presented as mean ± SE, ** *p* < 0.01; * *p* < 0.05.

**Figure 7 ijms-23-12278-f007:**
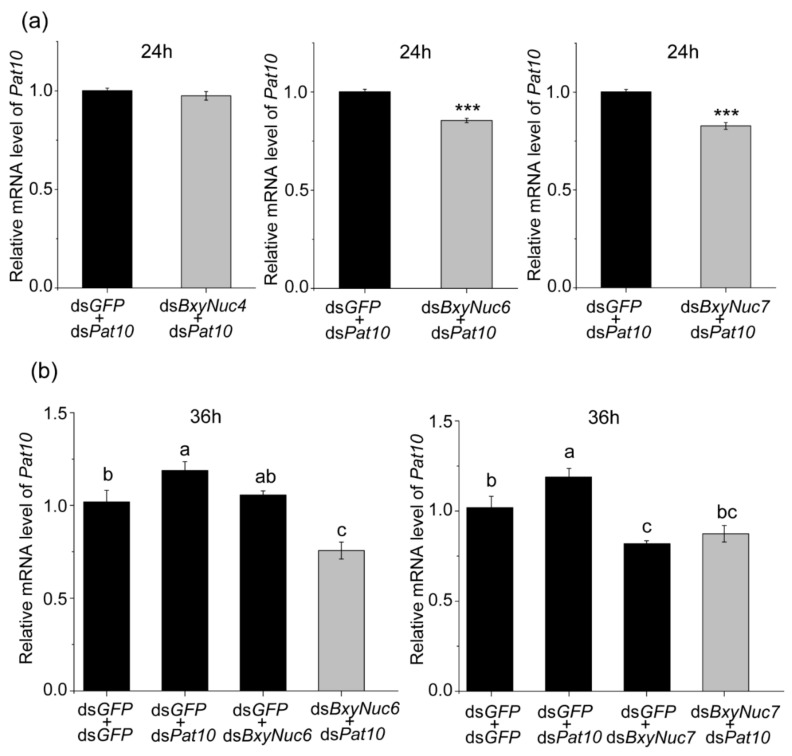
The relative expression level of *Pat10.* (**a**) The relative mRNA level of *Pat10* after interfering with *Pat10* gene and nucleases *BxyNuc4*/*BxyNuc6*/*BxyNuc8* genes simultaneously for 24 h. (**b**) The relative mRNA level of *Pat10* after interfering with *Pat10* gene and nucleases *BxyNuc6*/*BxyNuc7* genes simultaneously for 36 h. Gene expression differences between two groups were calculated using the independent samples *t* test. Data are presented as mean ± SE, *** *p* < 0.001; different letters indicate significant difference (*p* < 0.05).

**Table 1 ijms-23-12278-t001:** Information table of cloned *B. xylophilus* extracellular nucleases.

Name	Accession Number	ORF Length	Amino Acid Sequence Length	Signal Peptide	Theoretical pI/Mw
*BxyNuc1*	OP482151	993	330	20–21.VAS-KS	5.38/37,069.64
*BxyNuc2*	OP482152	1068	355	21–22.AEA-AG	6.37/40,630.98
*BxyNuc3*	OP482153	1035	344	20–21.VAA-SH	7.26/39,173.47
*BxyNuc4*	OP482154	933	310	17–18.GNA-AI	8.38/34,944.08
*BxyNuc5*	OP482155	890	296	15–16.TSA-QI	8.85/33,692.55
*BxyNuc6*	OP482156	927	308	18–19.ICG-TQ	9.12/35,178.94
*BxyNuc7*	OP482157	963	320	19–20.AQG-IG	5.72/36,009.80
*BxyNuc8*	OP482158	933	310	17–18.GNA-VI	8.58/34,493.57

## Data Availability

The nucleases *BxyNuc1-8* can be found using accession numbers OP482151-OP482158 from NCBI.

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
