# Peer review of "Identification of the Extracellular Nuclease Influencing Soaking RNA Interference Efficiency in Bursaphelenchus xylophilus"

_ijms, 2022, doi:10.3390/ijms232012278_

Round 1
Reviewer 1 Report
Dear Author(s),
Your manuscript is good, I appreciate your great work. Below are some suggested changes to improve it:
Line 34: Since you mention that it produces serious damage, you should add 1-2 more sources about the damage in recent years. The existing reference (from 982) is for the first report and not for the damages reported up to now. Be sure that these newly added sources will be detailed in the final list of References.
Line 49: Please make sure that it is written correctly 1 8 h or 18h?
Lines 72-76: In the Introduction, as a rule, the last sentence mentions what you want to do/analyze through your study. Therefore, the conclusion of the results should not be mentioned. Why? To make a smooth transition from what is already in the field and what you want to add to the current study.
As such, I suggest you write something like that: ...or another wording.
For Materials and Methods
At 4.1. Cultivation of nematodes
Lines 260-270: Please provide more details about the place/s from which you collected the nematodes, in response to the questions: How many nematodes were collected? Did you consider the tree or the surface of the forest/forest sector as a reference area? Also, do you place the experiment in time, in what period, in what year?
For Discussion
Lines 256-258: The conclusion is very brief. Since there is no separate chapter to highlight the relevant results, you should develop in several sentences what is essential. For example, you can move part of the final paragraph from the Introduction because you need to convince the reader more. Your work is great and deserves to be highlighted with more consistent conclusions.
For Refeerences
Line 370: Please add here and detail the 1-2 references chosen by you to reinforce the idea of damage from the Introduction.
Reviewer 2 Report
In general, the manuscript is well written and the results are interesting for publication.
Suggestions:
1-) In Introduction, describe more details about the symptoms of the disease and some biochemical features about the pathogen.
2-) What is the control method currently applied and its limitations? Or are the plants just eradicated? Please, explains more about this.
Minor corrections:
- Format "in vitro" to italic style (lines 72 and 111)
- Separate all reference numbers of words. Example: correct "...pine wilt disease[1]" to "...pine wilt disease [1]" (line 32)
